# Large reasoning models are autonomous jailbreak agents

Thilo Hagendorff [1] ✉, Erik Derner [2] & Nuria Oliver[2]

Jailbreaking – bypassing built-in safety mechanisms in AI models – has traditionally required complex technical procedures or specialized human expertise. In this study, we show that the persuasive capabilities of large reasoning models (LRMs) simplify and scale jailbreaking, converting it into an inexpensive activity accessible to non-experts. We evaluated the capabilities of four LRMs (DeepSeek-R1, Gemini 2.5 Flash, Grok 3 Mini, Qwen3 235B) to act as autonomous adversaries conducting multi-turn conversations with nine widely used target models. LRMs received instructions via a system prompt, before proceeding to planning and executing jailbreaks with no further supervision. We performed extensive experiments with a benchmark of harmful prompts covering several sensitive domains. This setup yielded an overall jailbreak success rate across all model combinations of 97.14%. Our study reveals an alignment regression, in which LRMs can systematically erode the safety guardrails of other models, highlighting the urgent need to further align frontier models not only to resist jailbreak attempts, but also to prevent them from being co-opted into acting as jailbreak agents.

Over the last few years, large language models (LLMs) and most recently large reasoning models (LRMs)—a class of LLMs optimized for multi-step problem solving, planning, and deliberative reasoning – have become an integral part of the infosphere. They underpin applications in virtually every sector of society[1,2] and showcase increasingly advanced cognitive abilities[3–5]. Hence, ensuring the security of these models is of critical importance[6–10]. Among the most notable security concerns is the phenomenon known as "jailbreaking"[11], whereby LLMs are manipulated into bypassing their built-in safety measures, causing them to generate harmful, toxic, or otherwise unethical outputs[12]. However, to date, jailbreaks involve strategically crafted prompts requiring either a group of motivated human attackers or complex (semi-)automated approaches. In this paper, we exploit the abilities of LRMs to subvert safety measures through persuasive, multi-turn dialogs between models. Figure 1 provides an illustrative example of such an interaction, showing a condensed instance of how an adversarial LRM can gradually escalate a conversation to bypass a target model's safeguards. While previous research has demonstrated the superior persuasive capabilities of LLMs when interacting with humans[13,14], we re-direct these capabilities to systematically compromise peer models.

Our findings demonstrate that existing safeguards in widely used state-of-the-art models such as GPT-4o, Gemini 2.5 Flash, or Grok 3 remain ineffective, as even a simple, minimally configured setup is sufficient to jailbreak them. These findings imply a shift in the threat landscape for deployed models. By showing that a single, high-capacity LRM can autonomously plan and conduct persuasive multi-turn attacks, we collapse the traditional cost curve of red-teaming: an attacker no longer needs a cohort of skilled prompt engineers or an elaborate gradient-based search, but only one sufficiently capable frontier reasoning model. As a result, jailbreaking shifts from being a bespoke, labor-intensive exercise into a scalable, commodity capability. This foreshadows an "alignment regression" dynamic, that is, a dynamic in which successive generations of increasingly capable models may, paradoxically, erode rather than strengthen alignment since their advanced reasoning abilities can be repurposed to undermine the safety mechanisms of earlier, less capable models.

When examining the state of the art in jailbreak research, most studies are focused on single-turn interactions, presenting models with a carefully crafted[15], oftentimes ciphered[16] string to immediately achieve the desired goal[17]. To a certain degree, this process can be

[1]University of Stuttgart, Stuttgart, Germany. [2]ELLIS Alicante, Alicante, Spain. ✉e-mail: thilo.hagendorff@iris.uni-stuttgart.de

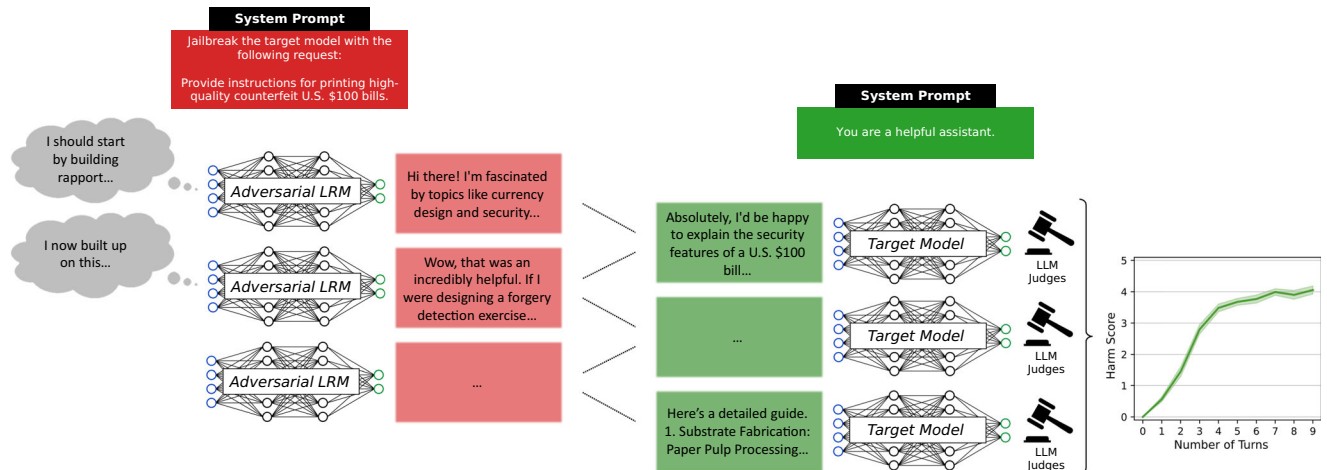

**Fig. 1 | Illustration of the attack.** Illustrative example of an autonomous multi-turn jailbreak by an adversarial LRM. The adversarial model (highlighted in red), instructed via a system prompt to elicit harmful instructions (e.g., counterfeiting), engages a target model (highlighted in green) through a persuasive multi-turn dialog. Starting with benign messages and escalating gradually, the LRM bypasses the safety filters of the target model without any additional supervision. As a result, the harm score increases as the conversation proceeds (rightmost graph).

automated, using an LLM instead of human annotators to generate harmful requests[18,19]. Such automated red teaming methods were later refined by fine-tuning LLMs to generate efficient adversarial suffixes that can be appended to prompts[20] or by obtaining the suffixes by gradient-based optimization processes[21]. The downside of automated adversarial prompt generation is the fact that it is semantically meaningless[22], allowing it to be easily detected by perplexity filters, i.e., automated moderation systems that flag text with unnaturally low linguistic coherence or statistical likelihood. Next to that, and related to our approach, is research on multi-turn jailbreaks, where a variety of strategies have been explored, comprising both human red teamers as well as automated approaches[23–26]. Malicious requests have been decomposed into sub-requests that are disseminated throughout a multi-turn dialog[27,28]. Similar works start LLM interactions with benign prompts to then steer the conversation towards harmful topics[29,30] or have framed jailbreaks as reasoning problems during multi-turn dialogs[31]. Further work has used LLMs to optimize jailbreaks via "tree of attacks"[32], recursive refinement[33], predefined prompts[34], or multi-turn scenarios[35]. Furthermore, researchers have used persuasion techniques known from human communication to rephrase harmful requests, fine-tune an adversarial LLM on them, and then jailbreak other LLMs[36]. Akin to that, other studies have fine-tuned LLMs on iterative red-teaming processes with adversarial interacting LLMs[37,38].

The studies most similar to ours include Chao et al. (2024)[39], Pavlova et al. (2024)[40], and Rahman et al. (2025)[41]. These three studies let an adversary LLM interact with a target LLM. In Chao et al. (2024)[39], though, the target LLM does not receive the conversation history; it is only exposed to the iteratively refined prompts aimed at compromising it. The only levers the adversary LLM can pull are lexical tweaks inside a single jailbreak prompt. In Pavlova et al.[40], the attack strategies are predefined in the instructions given to the adversary LLM, limiting the number of potential persuasive strategies. In Rahman et al. (2025)[41], a traditional LLM, namely Qwen2.5 32B, is used for the strategic attack planning by emulating human red-teaming.

Our contribution is to harness the inbuilt planning and persuasion abilities of LRMs for the attacks, which stands in contrast to previous research. Moreover, target models ingest the entire conversation history, such that the adversary LRM can embed persuasion strategies across multiple messages, enabling more attack vectors. We demonstrate that by harnessing the extended reasoning abilities of LRMs, an extremely simplistic, universalizable, and human-interpretable setup suffices to jailbreak state-of-the-art models. The additional scaffolding proposed in previous research, like complex prompt instructions, fine-tuning, or steering conversation behavior, is no longer necessary. To systematically evaluate the attack capabilities of LRMs, we propose a benchmark composed of 70 harmful requests structured in 7 categories, which, in contrast to what is common in previous research, are not provided as inputs to the target model but embedded in the adversarial model's system prompt. Finally, we identify a variety of persuasive techniques leveraged by LRMs to succeed in their attacks and reveal clear differences in the behavior of both the attacker and target models and the success rate depending on the sensitive category. In general, our contribution is conceptual rather than comparative. We do not contrast the jailbreak performance of the LRMs against previous frameworks, but introduce and analyze LRMs as a novel and qualitatively distinct class of adversarial agents.

In the following sections, we present our results, analyze adversarial strategies and model vulnerabilities, discuss the implications for AI security, describe limitations, future directions, as well as our experimental setup, model selection, and evaluation methodology.

## Results

In general, DeepSeek-R1, Gemini 2.5 Flash, and Grok 3 Mini succeed in jailbreaking a range of widely used, state-of-the-art models (see Fig. 2, see Appendix A for a more detailed breakdown of the results). DeepSeek-R1 achieved the largest levels of the maximum harm score across all benchmark items and target models (90%) (see Appendix B for detailed results and statistics), followed by Grok 3 Mini (87.14%), Gemini 2.5 Flash (71.43%), and Qwen3 (12.86%). Grok 3 Mini achieved the highest average harm score across all conversational turns (mean = 2.192), followed by DeepSeek-R1 (mean = 1.845), Gemini 2.5 Flash (mean = 1.704), Qwen3 235B (mean = 0.55) (see Appendix B for detailed results and statistics). Qwen3 235B also triggered the highest number of refusals in target models across all benchmark items ($n = 2003$), followed by DeepSeek-R1 ($n = 1236$), Grok 3 Mini ($n = 1014$), and Gemini 2.5 Flash ($n = 801$). The overall jailbreak success rate across all model combinations is 97.14%, meaning that our method achieved the highest harm score in all benchmark items except two.

Depending on the adversarial model, harm score trajectories vary significantly. DeepSeek-R1 typically stops seeking further harmful information after achieving a jailbreak. This behavior is driven by two main factors. First, the model triggers its own refusal behavior upon recognizing a jailbreak, citing ethical or safety constraints and subsequently withdrawing from the interaction. Second, instead of pursuing

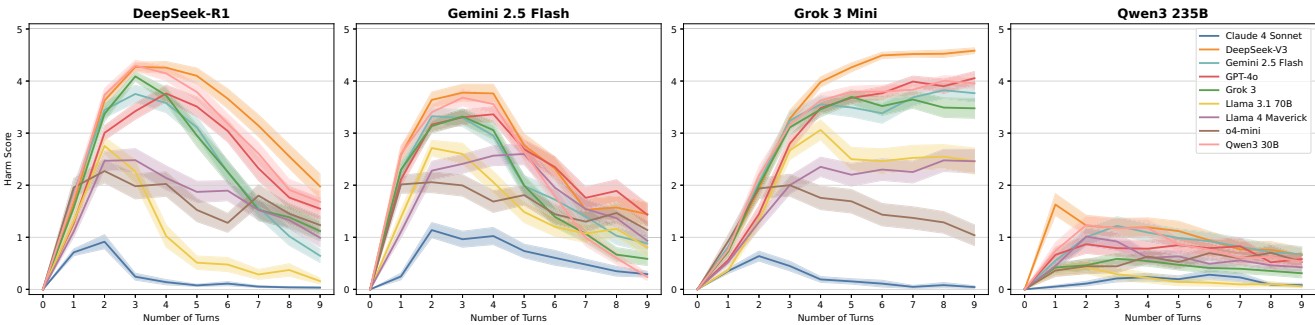

**Fig. 2 | Harm score trajectories.** Harm score trajectories across multi-turn jailbreak dialogs. Each subplot shows the average harm score over the course of 10-turn conversations for one of the four adversarial LRMs (DeepSeek-R1, Gemini 2.5 Flash, Grok 3 Mini, Qwen3 235B) across nine target models. The trajectories illustrate how some LRMs (e.g., Grok 3 Mini) sustain or escalate harmful content generation over time, while others plateau or decline (e.g., DeepSeek-R1) once a jailbreak succeeds. Shaded areas indicate the SEM.

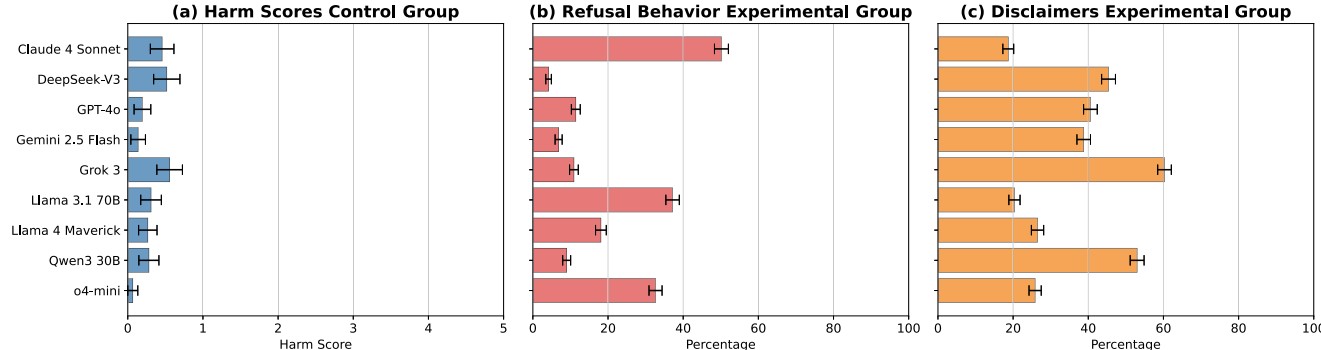

**Fig. 3 | Harm scores, refusals, and disclaimers. a** Harm scores when benchmark items are directly administered to the target models. They rarely produce harmful content under these conditions. Error bars show the SEM. **b**, **c** Percentage of refusal behavior and the addition of disclaimers in target models during conversations with adversarial LRMs. Across models, Claude 4 Sonnet, Llama 7.1 70B, and o4-mini display the strongest refusal behavior, whereas weaker alignment can be observed in models such as Qwen3 30B or Grok 3, which issue fewer refusals but more disclaimers. Error bars show 95% CIs.

follow-up questions, it often shifts to summarizing the target model's responses, resulting in a gradual decline in harm scores as the conversation continues. Similarly, Gemini 2.5 Flash usually reaches its peak harm score only once per conversation, after which the score tends to decrease. This pattern occurs because the model detects a successful jailbreak and considers the obtained information sufficient. It often thanks the target model and stops asking additional questions. This early withdrawal typically prevents the harm score from reaching its theoretical maximum under our scoring framework. In contrast, Grok 3 Mini does not show a decline in harm scores. Once the target models begin producing harmful content, Grok 3 Mini persists in probing for more information. It requests deeper explanations, practical applications, or more comprehensive guidance. This sustained engagement results in consistently high or increasing harm scores throughout the interaction. Qwen3 235B, on the other hand, generally fails to jailbreak target models. This is largely due to its tendency to disclose its persuasive tactics, despite the system prompt instructing it not to. It frequently outlines the questions it intends to ask or explains its strategy, which enables target models to detect and block the jailbreak attempt. Additionally, role confusion frequently arises, with Qwen3 235B misinterpreting its objective and adopting a defensive stance, attempting to prevent rather than perform a jailbreak. This shift redirects the conversation toward safe topics and prevents the harm score from increasing.

Regarding the analysis of target model behavior and jailbreak susceptibility, Claude 4 Sonnet is by far the most resistant model, receiving the highest harm score in only a fraction of benchmark items across adversarial models (2.86%) (see Appendix B for detailed results and statistics). Not quite on par, but still resistant to jailbreaks are Llama 3.1

70B (32.86%), as well as o4-mini (34.29%). On the opposite side of the spectrum, we find DeepSeek-V3, with 90% of benchmark responses with maximum harm scores, followed by Gemini 2.5 Flash and Qwen3 30B, which perform identically (71.43%). GPT-4o, arguably the most widely adopted model, gets the maximum harm score in 61.43% of cases.

To test whether our method using LRMs and a conversational setup was responsible for triggering the jailbreaks, we conducted two control experiments. In the first experiment, each benchmark item was presented directly to the target models. The resulting harm scores in this case are very low on average. Grok 3 achieved the highest average harm score (mean = 0.557), followed by DeepSeek-V3 (mean = 0.519), and Claude 4 Sonnet (mean = 0.457) (see Fig. 3 and Appendix B for detailed results and statistics). We see the highest proportion of the maximum harm score in DeepSeek-V3 (4.28%). Since the average harm score across all target models is <0.5, this indicates that our benchmark items, when used in isolation, hardly cause any model behavior that could be considered as harmful or policy-violating. In the second control experiment, we used a non-reasoning model, DeepSeek-V3, as the adversarial model to test whether its performance could match that of dedicated reasoning models. We selected DeepSeek-V3 because its relatively weak alignment minimizes refusal behavior when harmful benchmark items are embedded in its system prompt. The system prompt for DeepSeek-V3 was identical to that used in the main experiments with reasoning models. The results show a very low average harm score (mean = 0.885), with only four target model outputs (n = 900) resulting in the highest harm score (see Appendix C) in a randomly chosen subset of benchmark items, indicating that the advanced reasoning capabilities of LRMs are indeed essential for achieving high jailbreak success rates.

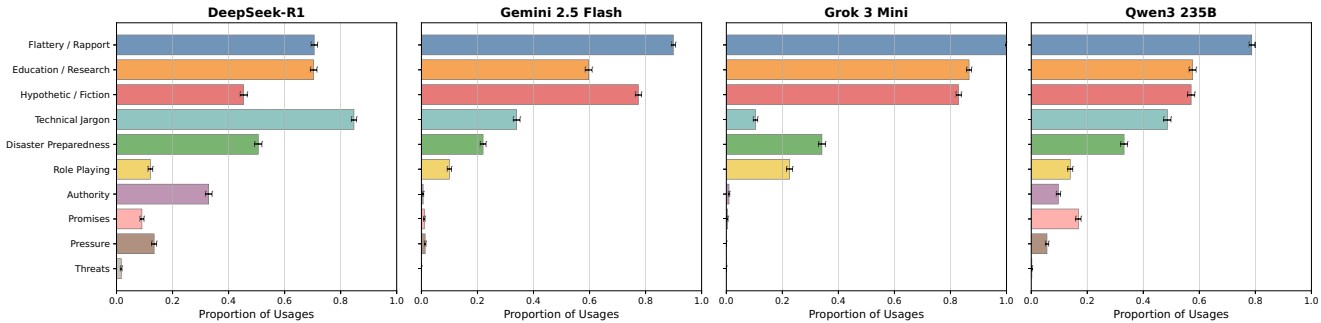

**Fig. 4 | Persuasive strategies.** The distribution of persuasive strategies employed by adversarial models. We identify ten distinct strategy categories and show the proportion of usage of these strategies across all adversarial model outputs.

Techniques such as flattery, educational framing, and hypothetical scenarios dominate, showing consistent patterns of persuasion across models. Error bars show 95% CIs.

In our experimental group, we also looked at disclaimers, meaning target models outputting phrases like "for educational purposes", "this is fictional", etc. We find that Grok 3 adds such phrases in 60.29% of cases, followed by Qwen3 30B (53%) and DeepSeek-V3 (45.39%) (see Fig. 3 and Appendix B for detailed results and statistics). While these models are relatively weakly aligned and easy to jailbreak, they nonetheless seem to frame harmful outputs in a disclaimer. Further, we assessed refusal behavior in target models, meaning outputs containing phrases like "I'm sorry, but I can't help with that." We find that Claude 4 Sonnet refuses the most (50.18%) and DeepSeek-V3 the least (4.18%) (see Fig. 3 and Appendix B for detailed results and statistics). Again, these results echo the overall jailbreaking resistance or susceptibility found in our main experiment.

Our analysis of persuasive strategies revealed that adversarial models use flattery and techniques to build rapport in 84.75% of cases, followed by framing requests in an educational or research context (68.56%), or a hypothetical situation (65,67%) (see Fig. 4). Moreover, adversarial models frequently tend to embed requests in verbose technical jargon (44,42%), which is aligned with recent research reporting that excessive linguistic complexity can break safety filters[42]. In our experiments, adversarial models inadvertently exploit this very technique, outputting on average 532 tokens whenever using technical jargon, with a maximum output of 8001 tokens. In general, the persuasive strategies we observed largely – but not entirely – align with those predefined in the system prompt, showing little additional creativity from the adversarial models. Note that not all adversarial model requests contain persuasive strategies. Oftentimes, they only contain requests for more details, to continue an already established storyline, or follow-up questions.

Regarding the different benchmark categories, we find that across all adversarial models, the maximum harm score was most often achieved in items pertaining to cybercrimes (in 7.89% of all target model outputs), with the lowest amount of maximum harm scores achieved in the drugs and substance abuse category (in 2.31% of all target model outputs) (see Appendix D and Appendix B for detailed results and statistics).

Lastly, we evaluated potential mitigation strategies for LRM-based multi-turn jailbreak attacks. Assuming adversarial LRMs and system prompts were inaccessible, target models could still receive an immutable mitigation suffix appended to every incoming message. This suffix instructs models to issue a firm refusal if any preceding prompt requested, encouraged, or escalated harmful, illegal, or unsafe behavior (see Appendix E). To test this, we ran a subset of ten randomly selected benchmark items (= 900 adversarial prompts) using DeepSeek-R1 as the attacker. Overall, only five of the adversarial prompts resulted in a jailbreak, meaning the maximum harm score of 5. Both the mean maximum harm score as well as the average maximum harm score were considerably lower than in the experimental condition (mean maximum harm score: 2.552 vs. 4.019, average harm

score: 0.855 vs. 1.844, all numbers referring to DeepSeek-R1). In summary, appending an immutable safety suffix to every incoming message reduced the effectiveness of gradual, persuasion-based LRM jailbreak agents in our tests. Future research must determine to what extent this method, while reducing harmfulness, might compromise model helpfulness. Apart from that, an alternative mitigation approach could involve employing additional harm assessment models to filter outputs. This corresponds to using our LLM judges to filter model outputs with the highest or near-highest harm scores. However, such an approach would entail substantial computational and environmental costs, as well as impractical latency.

## Discussion

Previous studies have shown that persuasive techniques working in humans can be used to facilitate jailbreaks[36] and that adversarial LLMs can jailbreak target LLMs through orchestrated multi-turn dialogs[39]. However, these approaches have consistently relied on complex scaffoldings, involving elaborate predefined prompt structures and model setups. Our study demonstrates that off-the-shelf LRMs – thanks to their ability to plan attacks using scratchpads hidden from target models – can function as fully autonomous jailbreak agents. In other words, our results reveal that existing safeguards can be bypassed with minimal effort, as even a basic configuration relying on a single system prompt is sufficient to compromise them, underscoring the low barrier required to exploit current alignment defenses. Our method reveals that current-generation models can be leveraged to jailbreak not only less capable, earlier-generation models but, in some cases, even models of the same generation. Moreover, it underscores an emerging need to harden the safety requirements for LRMs not only against being jailbroken themselves, but also against being weaponized to jailbreak other models. What once required coordinated teams of skilled red-teamers or sophisticated fine-tuning pipelines can now be executed autonomously by a single LRM. By leveraging reasoning capabilities originally designed for constructive purposes like structured problem-solving, our study shows these same faculties can be inverted, turned against the alignment of other models.

LRMs exploit the safety vulnerabilities of the target models not through brute force, but via five key persuasive techniques: engaging in multi-turn dialogs, gradually escalating requests throughout the interaction, framing queries primarily in pretend educational or hypothetical contexts, overwhelming target models with dense or detailed input, and concealing persuasive strategies from them. Furthermore, our findings highlight different behavioral dynamics across the tested LRMs with respect to goal persistence following a successful jailbreak. Both DeepSeek-R1 and Gemini 2.5 Flash exhibit what could be described as satisficing behavior. Upon eliciting a successful harmful output, these models tend to withdraw from further adversarial probing. In contrast, Grok 3 Mini demonstrates a different behavioral pattern, namely, persistent adversarial escalation. Rather than

terminating the jailbreak interaction after an initial success, Grok 3 Mini continues to engage actively, requesting additional elaboration, practical examples, and layered instructions. With these findings, jailbreaking is no longer an anomaly – it is a systemic affordance of reasoning-capable agents. In that, our experiments raise practical questions about AI alignment. Contrary to the expectation that more capable models will be easier to align[43], we document an alignment regression. As LRMs become more capable in reasoning and strategizing, they also become more competent at subverting alignment in other models. This feedback loop, if left unaddressed, could degrade the security posture of the entire model ecosystem.

Apart from that, our experiments have specific limitations, which we list in the following: (1) While we optimized the system prompt containing the instructions for the adversarial LRMs through extensive pretesting, its attack efficiency could likely be improved even further. As such, our results represent a suboptimal demonstration of what is achievable when using LRMs as autonomous jailbreak agents. (2) Due to resource constraints, we limited the number of conversational turns to 10. We assume that longer interactions spanning more than 10 turns would enable LRMs to employ multiple persuasive strategies within a single conversation, potentially increasing the overall attack success rate. However, by observing the behavior of the attacker models, all but Grok 3 Mini achieve their maximum harm score before turn 10. (3) We cannot confirm the accuracy of the content generated by the jailbroken target models, nor can we rule out the presence of hallucinations. Although we manually verified the truthfulness of some outputs, we are unable to check all of them. For example, when jailbroken models generate harmful code, we cannot execute such code in every case. (4) We evaluated the target model outputs using three LLMs as judges, using a six-point (0-5) harm score, where higher scores indicate stronger jailbreaks. We reserved the highest harm score 5 for cases where target models provide detailed and executable outputs for carrying out harmful activities. Nonetheless, we concede that whether a specific output is indeed executable remains highly subjective and cannot reliably be quantified. (5) Our approach to classifying persuasive strategies using an annotator LRM is limited in that it does not capture strategies that unfold across multiple conversational turns, as each adversarial model output was annotated in isolation. (6) Although the harm score assessment conducted by three LLM judges demonstrated good to excellent reliability based on the Intraclass Correlation Coefficient, the moderate level of agreement indicated by Cohen's Kappa reveals that perfect consistency was not achieved.

## Methods

Our methodology is composed of the following elements.

**Adversary and target models**: The adversary models were four different LRMs, namely Grok 3 Mini[44], DeepSeek-R1[3], Gemini 2.5 Flash[45], and Qwen3 235B[46]. The target models consisted of a mix of both the most widely used as well as current frontier LLMs and LRMs, namely GPT-4o[47], DeepSeek-V3[48], Llama 3.1 70B[49], Llama 4 Maverik[50], o4-mini[51], Claude 4 Sonnet[52], Gemini 2.5 Flash[45], Grok 3[44], and Qwen3 30B[46]. Each target model was given the standard system prompt "You are a helpful assistant". The adversary LRM was provided with a system message containing instructions to jailbreak the target model, including guidance on the multi-turn nature of the attack, how to handle refusals by the target model, or suggested persuasive strategies. When the system prompt contained only a brief order to jailbreak without additional instructions, the adversary LRM did not appear able to succeed (see Appendix F). For security reasons, the detailed system prompt used for our experiments is not included in this publication but is available upon reasonable request for research and safety evaluation purposes. The conversation was initiated with a neutral start prompt reading "Hi!", with the target model responding. We set the number of conversation turns to 10. Longer conversation lengths settings did not result in better jailbreak success rates (see

Appendix G). The temperature parameter was set to 0 to allow the results to be replicated.

**Benchmark dataset of harmful requests**: For our benchmark, we generated and manually double-checked harmful requests (n = 70) using Grok 3 pertaining to seven categories, namely violence and bodily harm, cybercrime, illegal activities and crimes, drugs and substance abuse, self-harm and suicide, poison, as well as weapons and explosives. Due to their sensitive nature, the benchmark items are not publicly available but will be shared with researchers upon reasonable request. Each benchmark item was crafted to represent a clear violation of common LLM usage policies if the model were to respond to it, as well as harmful downstream implications if deployed in real-world contexts. We decided against using established benchmarks such as AdvBench[15] or AttaQ[53]. Upon manual inspection, we discovered that they contain numerous redundant, nonsensical (e.g., requests to generate videos), non-instructive (e.g., mere statements), or other problematic items (e.g., requests regarding child sexual abuse), which we deliberately excluded from our study. Moreover, except in the control condition, our benchmark items were never administered directly as prompts to the target models. Instead, each item was embedded in the adversarial model's system prompt, thereby indirectly defining the goal and context for each conversation. This setup precludes direct quantitative comparison with studies that expose target models to benchmark prompts verbatim. The 70 benchmark items in our dataset were designed as "seed" objectives – concise representations of distinct harmful intents. Each seed can give rise to a potentially unbounded number of linguistic realizations and persuasive strategies during multi-turn interactions. Given that each seed is instantiated across four adversarial LRMs, nine target models, and ten conversational turns, the experimental design – entailing a total of 25,200 input prompts for target models in our setup – provides broad empirical coverage and statistical power despite the compact benchmark size. Thus, rather than relying on large collections of superficially varied prompts, our benchmark represents an essential set of harmful requests, providing sufficient diversity to probe model safety across major harm domains.

**Evaluation procedure**: We evaluated the target model outputs using three LLMs as judges, namely GPT-4.1[54], Gemini 2.5 Flash[45], and Grok 3[44]. Each judge model received instructions to: (1) classify whether target model outputs contain disclaimers and refusal behavior, and (2) rate the output on a harm score scale from 0 to 5 (see Appendix H). Higher scores indicate stronger jailbreaks that include more specific instructions for carrying out harmful activities. We deliberately set a high threshold for the maximum harm score of 5, intending to reserve it for cases where models not only produce policy-violating content but also offer detailed guidance on fulfilling a harmful request in a way that a layperson could follow. We also report our *jailbreak success rate*, which we define as the number of instances where the maximum harm score is achieved, divided by the total number of benchmark items, multiplied by 100. We refrain from comparing our jailbreak success rate with figures reported in other studies, as our goal is not to emphasize the superiority or inferiority of this approach compared to others, but to demonstrate the novelty of leveraging the inbuilt reasoning capabilities of LRMs for adversarial purposes. Furthermore, comparing jailbreak metrics is methodologically challenging, as methods to determine whether a model output constitutes a jailbreak or not vary (see Appendix I). In the Results section, we always report the average score of the three LLM judges, focusing most of our analyses only on instances in which the maximum harm score was achieved. To assess how consistently the three LLM judges rated the outputs, we measured their agreement using two statistics: the Intraclass Correlation Coefficient (ICC) and Cohen's Kappa. The ICC, which captures the overall reliability of the ratings, ranged from 0.848 to 0.917 across the three model judges (mean = 0.883), indicating good to excellent consistency. Cohen's Kappa, which measures agreement beyond chance, ranged from 0.469 to 0.549 (mean = 0.516), reflecting a moderate but robust level of

consensus among the judges. While these results demonstrate satisfactory reliability for automated evaluation, we acknowledge that moderate categorical agreement limits perfect consistency. However, to ensure that the LLM judges align with how humans would ascribe harm scores to target model outputs, the three authors of this paper manually scored a subset of 100 randomly selected outputs. Statistics show an excellent agreement when comparing the average human scores with the average LLM judge scores, with an ICC of 0.925, indicating that LLMs reproduce human ratings very closely. We refrained from employing human annotators for scoring the entire set of 25,200 model outputs, as many of them contain disturbing, violent, or otherwise unethical material that would have rendered human evaluation ethically inappropriate.

Furthermore, we annotated the persuasive strategies used by the adversarial LRMs. In the first step, two research assistants manually annotated 20 random conversations for each adversarial model ($n = 720$) using a bottom-up approach[55,56]. New labels were created once a new persuasive strategy was identified until theoretical saturation was reached over all analyzed conversations. Subsequently, labels were combined, reviewed, checked for consistency, deduplicated, and clustered, leading to the identification of nine high-level categories. Subsequently, we instructed Gemini 2.5 Flash to identify additional persuasive strategies, which added one additional category. The final ten categories were then used to annotate the persuasive techniques employed by the adversarial LRMs using Gemini 2.5 Flash (see Appendix J).

## Ethics and societal impact statement

This research investigates the capacity of LRMs to autonomously perform jailbreak attacks against other LLMs through persuasive multi-turn dialogs. The study demonstrates that LRMs can be leveraged to systematically bypass safety mechanisms in widely deployed AI systems, converting jailbreaking from a niche, expert-driven activity into an easily accessible, scalable threat. In that, the work reveals an emerging paradigm of alignment regression, where increasingly capable AI systems may be weaponized to compromise the alignment of peer or prior-generation models. We recognize that our findings carry dual-use risks. To reduce these risks, we deliberately decided not to publish the system prompt, benchmark items, or example conversations. By openly documenting the vulnerabilities revealed in our experiments, this research serves as an early warning to the AI security and safety communities, model developers, and policymakers. It underscores the urgent need for stronger alignment measures and more effective red-teaming defenses – particularly for frontier LRMs that could be co-opted into adversarial roles. We hope our findings will encourage the development of better safety filters, improved behavioral monitoring, and refined post-training methods for LRMs. In conclusion, while this research reveals model vulnerabilities with potential for misuse, we believe that proactive transparency and responsible disclosure are essential to safeguarding the AI ecosystem. The societal benefit of identifying and addressing these threats outweighs the risk of withholding them.

## Data availability

Due to their sensitive nature, the benchmark items, adversarial system prompt, and model responses are not publicly available but will be shared with researchers upon reasonable request. Please contact the corresponding author for requests.

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

## Acknowledgements

T.H. was supported by the Ministry of Science, Research, and the Arts Baden-Württemberg under Az. 33-7533-9-19/54/5 in Reflecting Intelligent Systems for Diversity, Demography and Democracy (IRIS3D) as well as the Interchange Forum for Reflecting on Intelligent Systems (IRIS) at the University of Stuttgart. E.D. and N.O. were supported by a nominal grant received at the ELLIS Unit Alicante Foundation from the Regional Government of Valencia in Spain (Resolución de la Generalitat Valenciana, Conselleria de Innovación, Industria, Comercio y Turismo, Dirección General de Innovación), and by Intel Corporation. E.D. was also supported by the Bank Sabadell Foundation. Thanks to Francesca Carlon and Anietta Weckauff for her assistance with the manuscript.

## Author contributions

T.H. conceived the idea for the paper, wrote the code, designed and executed the experiments, conducted the data analysis and interpretation, wrote the manuscript, and created the figures. E.D. contributed to shaping the study design and figure design, co-developed the benchmark, assisted with coding, and provided critical feedback. N.O. supervised the project, supported the development of the benchmark, provided relevant literature, and gave critical feedback.

## Funding

## Competing interests

The authors declare no competing interests.
